# Numerical Investigation of Asphalt Concrete Fracture Based on Heterogeneous Structure and Cohesive Zone Model

**Jiaqi Chen, Xu Ouyang and Xiao Sun \***

Department of Civil Engineering, Central South University, Changsha 410075, China
\* Correspondence: sunx@csu.edu.cn

**Abstract:** The fracture behavior of asphalt concrete is closely related to its internal structure. A deep understanding of the relationship between the internal structure and fracture behavior of asphalt concrete is very important for sustainable and durable pavement design. In this paper, a CZM-based FE model was developed to investigate the fracture behavior of asphalt concrete. An image-aided approach was used to generate the 3-D internal heterogeneous structure of asphalt concrete. A series of 2-D cross sections were extracted from the 3-D structure for finite element modeling. Then numerical simulations of SCB tests were conducted and validated with experimental results. With the validated CZM-based FE model, the effects of some critical factors, including temperature, loading rate, aggregate geometry, fracture strength, and fracture energy, on the fracture behavior of asphalt concrete were investigated. The analysis results showed that the average damage of the adhesive elements was higher than that of the cohesive elements at the peak load. At lower temperatures, asphalt concrete tends to crack earlier, and the cracking path tends to be marginally closer to the aggregates. A higher loading rate may induce more, but minor, element damage since the CZM elements in asphalt mortar cannot bear much more stress through deformation. Angular aggregates may induce a higher percentage of damaged elements, especially adhesive-damaged elements. On average, each 10% increase in fracture energy allows the specimen to bear 2.31% more load and 2.82% more displacement. Sufficient fracture energy could improve the ability of asphalt concrete to resist fracture.

**Keywords:** asphalt concrete fracture; cohesive zone model; finite element; semi-circular bending test; adhesive damage; cohesive damage

## 1. Introduction

Pavement cracking is one of the common damages in asphalt pavements, which has an adverse effect on pavement durability. Visible cracks in the pavement structure usually initiate from local microcracks in the asphalt concrete. Since asphalt concrete is a composite heterogeneous material composed of asphalt mastic and mineral aggregates, the fracture behavior of asphalt concrete is affected by its internal composition and meso-structure [1–3]. A deep understanding of the relationship between the internal structure and fracture behavior of asphalt concrete is very important for sustainable and durable pavement design.

In order to study the fracture behavior of asphalt concrete, numerous experimental studies have been carried out from a macro perspective. The three-point bending test is among the most commonly used test methods for fracture evaluation [4,5]. Studies have shown that the mechanical behavior of asphalt concrete during pure mode I fracture and pure mode II fracture is well reflected through semi-circular bending (SCB) tests [6]. In addition, the SCB tests were also used to evaluate the fracture behavior of asphalt concrete containing recycled aggregates [7]. Although these experiments could provide the homogenized macroscopic mechanical properties of asphalt concrete, it is not easy for these experiments to reveal the internal damage mechanism of asphalt concrete from a meso or

microscope. In order to further study the internal fracture behavior of asphalt concrete, experiments focusing on the local structures of asphalt concrete have been conducted. For example, pull-off tests and direct tensile tests have been used to study the fracture behavior of the interface between aggregate and asphalt mortar [8–12]. Table 1 offers a brief review of some of the experimental standards. These experiments provide quantitative indicators for evaluating the fracture behavior of asphalt concrete, but it is difficult for these experiments to identify crack initiation and propagation. In this case, the numerical models which involve the heterogeneous internal structure of asphalt concrete are more applicable.

**Table 1.** Summary of different experimental standards.

| Method | Semi-Circular Bending | Pull-Off Tests | Direct Tensile Tests |
|---|---|---|---|
| Parameters | Fracture energy Fracture toughness Critical strain energy J-integra | Pull-off tensile strength Burst pressure Contact area of gasket with reaction plate Area of pull-off stub | Tensile strength Tensile stress Strain energy density Effective gauge length |
| Limitation | Only macro parameters can be explored | Complex experiments and demanding experimental setup | High requirements for experimental setup and operation |
| Specifications | [13–16] | [17] | [18–20] |

These numerical models are usually based on the extended finite element method (XFEM) or the cohesive zone model (CZM). Since the XFEM does not require special elements embedded at the element interface, the crack propagation is relatively flexible for XFEM, and this method has been widely used to investigate the fracture behavior of asphalt concrete [21–25]. Due to the heterogeneous internal structure, the damage evolution of asphalt concrete is usually complex, which includes adhesive damage at the aggregate–asphalt interface and cohesive damage in asphalt mortar [26–28]. However, it is difficult for XFEM to simulate the interaction between the cracks in asphalt and in the aggregate–asphalt interface [29]. Alternatively, CZM provides a convenient way for simulating damage and fracture evolution both at the interface and in asphalt mortar [30–33]. The adhesion damage is usually achieved by introducing cohesive elements at the aggregate–asphalt interface [34–36]. In order to avoid excessive computational cost, 2-D CZM analysis is usually used. The microstructure of asphalt concrete, such as aggregate shape and distribution, has an important influence on macroscopic properties [37–40]. In order to reflect the microstructure of asphalt concrete in numerical models, numerical image identification and parametric modeling are commonly used. Numerical image identification can transform high-precision images of asphalt concrete into geometric models that characterize the microstructure of asphalt concrete [41–43]. However, this process often requires operational skills and expensive experimental equipment [30,41]. Although the parametric modeling of 2-D asphalt concrete is technically feasible [44], studies show that the 2-D asphalt concrete specimens generated with the 3-D volumetric parameters are usually different from the 2-D cross sections cut from the real 3-D specimens [45]. This problem can be solved by converting 3-D volumetric parameters to 2-D area parameters [46], but the conversion criteria are not uniform. Therefore, in this paper, a 3-D model of asphalt concrete was built considering the real aggregate shape first. Then a series of two-dimensional models were cut from a 3-D model to ensure the reality of the microstructure for 2-D models.

## 2. Objective

The primary objective of this study is to develop a CZM-based FE model to evaluate the fracture behavior of asphalt concrete. An image-aided approach was used to generate the 3-D internal heterogeneous structure of asphalt concrete. A series of 2-D cross sections were

extracted from the 3-D structure for finite element modeling. Then numerical simulations of SCB tests were conducted and validated with experiment. With the validated CZM-based FE model, the effects of some critical factors, including temperature, loading rate, aggregate geometry, fracture strength, and fracture energy, on the fracture behavior of asphalt concrete were investigated.

## 3. Development of CZM-Based FE Model

### 3.1. Modeling 2-D Virtual Specimen of Asphalt Concrete

In this paper, 2-D virtual specimens of asphalt concrete were used for numerical analysis to ensure computational efficiency. It has been found that 2-D asphalt concrete specimens generated from 3-D volumetric parameters are usually different from the 2-D cross sections cut from real 3-D specimens [45]. Although this problem can be solved by converting 3-D volumetric parameters to 2-D area parameters, there is still uncertainty in the conversion criteria. Therefore, in this paper, the 3-D virtual specimen of asphalt concrete was generated first with an image-aided approach presented previously [47]. Then a series of 2-D cross sections were extracted from the 3-D virtual specimen and were used for further finite element analysis. In this way, the generated 2-D virtual specimens could properly represent the internal structures of asphalt concrete without considering the complex conversion between 2-D and 3-D parameters. The general process for the above modeling method can be summarized as follows:

(1) Scan the real aggregate particles to obtain the 2-D geometries of aggregates in the format of a binary image. Then convert the binary images to closed polygons.

(2) Generate random points inside the 2-D polygons and assign random spatial coordinates to these points. Every three points could form a triangle. A closed polyhedron that represents the 3-D virtual aggregate could be generated by a series of triangles in the format of a stereolithography (STL) file.

(3) Import the 3-D virtual aggregates into PFC software. With the assistance of PFC software, the aggregates were placed into a cylinder space based on the prescribed material composition without overlapping with each other. By this step, the 3-D virtual specimen of asphalt concrete has been developed. In this paper, the generated 3-D virtual specimen is a cylinder with 150 mm in diameter and 150 mm in height.

(4) A series of 2-D cross sections were extracted from the 3-D virtual specimen. The aggregate contours (polygons) in each cross section were converted into a Drawing Exchange Format (DXF) file. These DXF files were imported into ABAQUS software to develop 2-D finite element models.

The schematic diagram of the above process is shown in Figure 1.

### 3.2. Cohesive Zone Model

The cohesion of the asphalt mortar and adhesion between aggregate and asphalt mortar are critical material properties for the fracture behavior of asphalt concrete [48]. In this paper, it is assumed that cracks only initiate and extend inside the asphalt mortar and aggregate-asphalt mortar interface. This is because the strength of aggregate is usually much higher than that of asphalt mortar. The fracture of aggregate is only observed when the temperature is very low, such as below $-20\ ^\circ$C [30], which is beyond the scope of this paper. The cohesive zone model (CZM) was used in this paper for simulating the fracture behavior of asphalt concrete. In the CZM, adhesion damage and cohesion damage were characterized by inserting zero-thickness cohesive elements into the aggregate–asphalt mortar interface and the asphalt mortar, respectively (Figure 2). The bilinear traction–separation law was applied to the cohesive elements to achieve damage variation and crack expansion [49], as shown in Figure 3. The bilinear traction–separation law is mainly divided into the elastic phase before the peak load and the softening phase after the peak load. After the cohesive elements reach maximum traction stress under external action, it enters the softening phase, and damage begins to occur. In this stage, the stiffness of the cohesive element gradually decreases with the external action. Until the threshold value

(fracture displacement or fracture energy) is reached, the traction decreases to zero, the two contact surfaces are completely separated, the element fails completely, and cracks are caused. As the number of complete failures of cohesive elements increases, the length of cracks also increases.

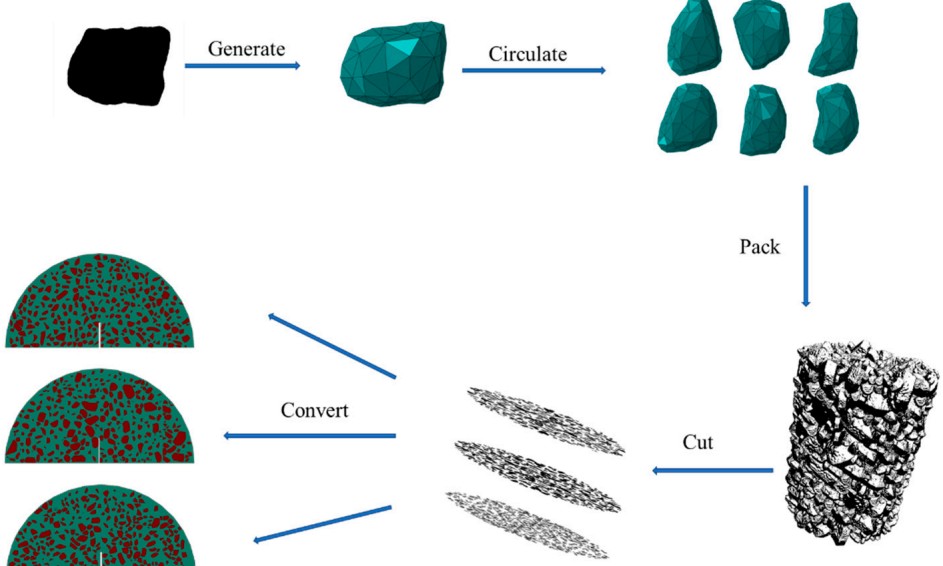

**Figure 1.** Schematic diagram of the building of the heterogeneous model.

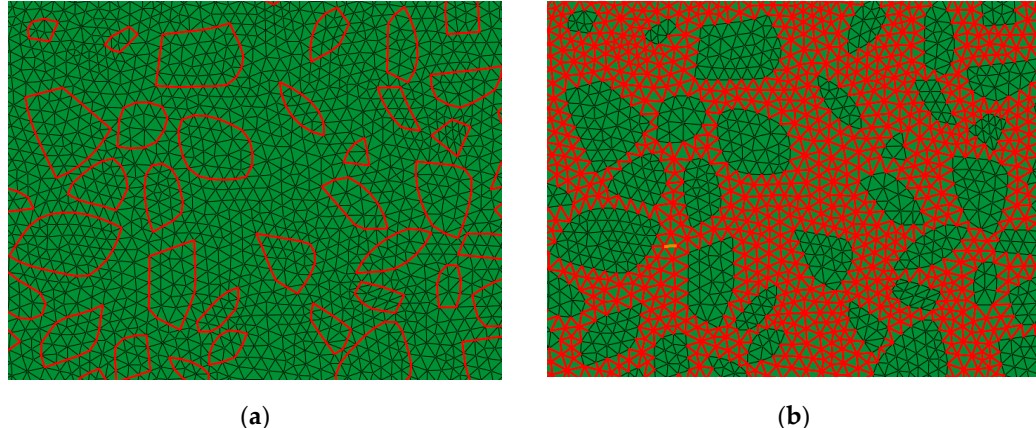

(**a**)　　　　　　　　　　　　　　　　　　　　　　　　(**b**)

**Figure 2.** Cohesive elements (**a**) at aggregate–asphalt interface for adhesion; (**b**) in the mortar for cohesion.

### 3.3. Definition of Materials

In this paper, asphalt concrete is regarded as a composite material consisting of discrete coarse aggregates (particle size > 2.36 mm) and a continuous matrix consisting of asphalt mortar, air voids, and fine aggregates (particle size < 2.36 mm). The aggregate gradation is shown in Table 2. The air void content was 6%, and the asphalt content was 5%. The aggregates were defined as a linear elastic material, while the asphalt mortar was defined as a viscoelastic material. The loading rate Maxwell model shown in Figure 4 was used to characterize its behavior, and the detailed parameters in the format of the Prony series are shown in Table 3 [30]. Parameters for the CZM model used in this paper were taken from existing literature, as shown in Table 4 [30,36].

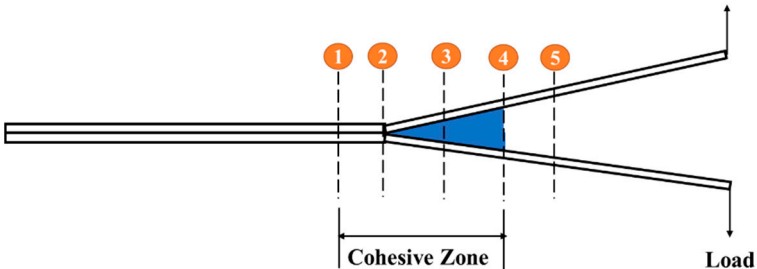

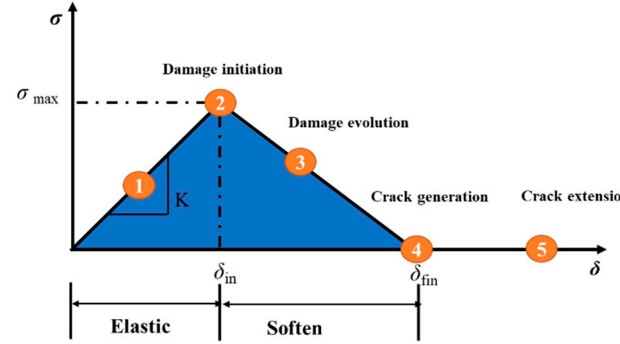

**Figure 3.** The bilinear traction–separation law.

**Table 2.** Aggregate gradation.

| Sieve Size (mm) | 16.0 | 13.2 | 9.5 | 4.75 | 2.36 | 1.18 | 0.6 | 0.3 | 0.15 | 0.07 |
|---|---|---|---|---|---|---|---|---|---|---|
| Passing percentage (%) | 100 | 96.0 | 82.1 | 52.2 | 30.9 | 23.0 | 16.9 | 11.0 | 8.4 | 6.8 |

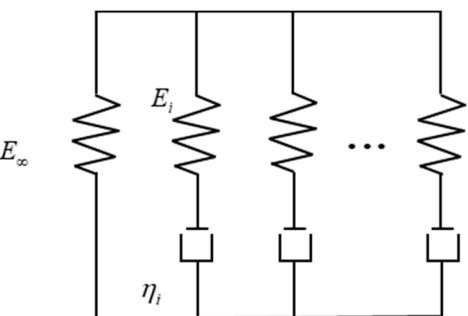

**Figure 4.** Maxwell model.

**Table 3.** Parameters for the Prony series [30].

| Temperature (°C) | $g1$ | $\tau1$ | $g2$ | $\tau2$ |
|---|---|---|---|---|
| −10 | 0.08586 | 5.1058 | 0.84331 | 74.371 |
| 0 | 0.11909 | 4.53336 | 0.86705 | 54.604 |

**Table 4.** Parameters for CZM [30,36].

| Temperature (°C) | Phase | $E$ (MPa) | Poisson's Ratio | $\sigma$ (MPa) | $G_I$ (J/m²) | $G_I$ (N/mm) |
|---|---|---|---|---|---|---|
| −10 | Aggregate | 55,500 | 0.15 | / | / | / |
| | Mastic | 805.6 | 0.25 | 4.35 | 805 | 0.805 |
| | Interface | 805.6 | 0.25 | 3.92 | 403 | 0.403 |
| 0 | Aggregate | 55,500 | 0.15 | / | / | / |
| | Mastic | 621.6 | 0.25 | 3.41 | 950 | 0.950 |
| | Interface | 621.6 | 0.25 | 3.11 | 475 | 0.475 |

*3.4. FE Model Paraments*

In total, three 2-D virtual specimens of asphalt concrete were generated and imported into ABAQUS for FE analysis. An example of the 2-D FE model is shown in Figure 5. The diameter of the specimen was 150 mm, and the notch depth was 20 mm. The element type for the aggregates and asphalt mortar was selected as CPS3 (3-node linear plane stress triangle), while the element type for the cohesive elements was selected as COH2D4 (4-node two-dimensional cohesive element). The number of elements in the model varied due to the number and size of aggregates in each virtual specimen. More specifically, the maximum number of elements is about 51,000, and the minimum number is about 44,000. The distance between the two supports at the bottom of the specimen was 120 mm, namely, 0.8 times the diameter. The two supports were set to be rigid, in contact with the model, with 0 degrees of freedom. The contact between the SCB specimen and the supports was face-to-face contact. The normal contact behavior was defined by „Hard Contact", while the tangential contact behavior was defined by the friction model. A vertical downward loading was applied to the virtual specimen through a rigid indenter with a loading rate of 5 mm/min.

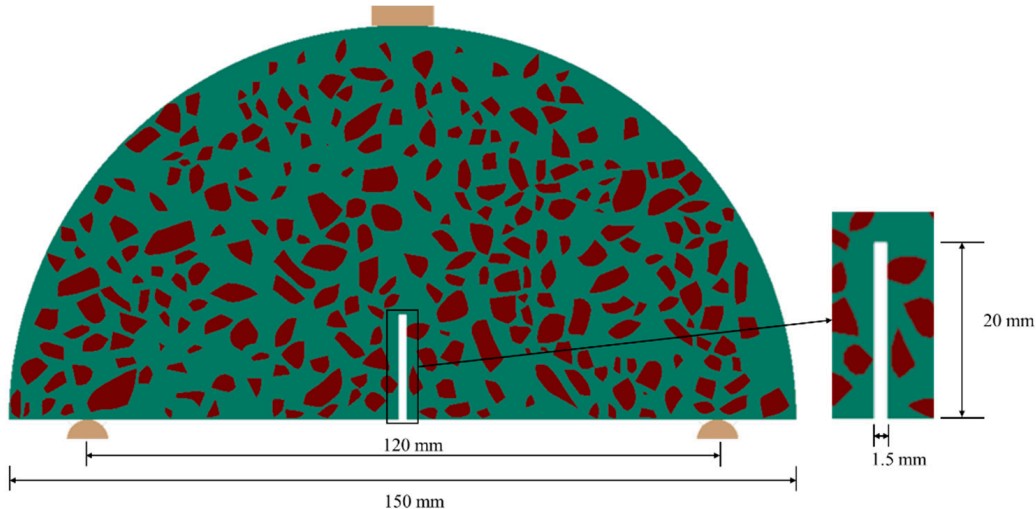

**Figure 5.** Example of the 2-D FE model.

## 4. Experiment and Model Validation

### 4.1. Experimental Setup

Semi-circular bending (SCB) tests were conducted at $-10\ ^\circ$C and $0\ ^\circ$C to investigate the fracture behavior of asphalt concrete and validate the CZM-based FE model. The aggregate gradation used in the experiments was consistent with that used in the FE model. Cylindrical specimens with a height of 150 mm and a diameter of 150 mm were compacted using a Superpave rotary compactor (SGC) under a compaction pressure of 600 kPa. The SCB specimens were drilled and cut from the central part of the cylindrical specimens to avoid uneven porosity. The specimens used in the SCB tests were 150 mm in diameter and 25 mm in thickness. A notch with 20 mm in depth was precut, as shown in Figure 6.

The SCB tests were conducted by a multi-functional testing machine (UTM-250) with an ambient chamber. For each test, the specimen was pre-placed in the ambient chamber for more than 4 h. The temperature in the ambient chamber was set to be constant ($-10\ ^\circ$C or $0\ ^\circ$C) during the test. The vertical loading was applied with a loading rate of 5 mm/min for each test.

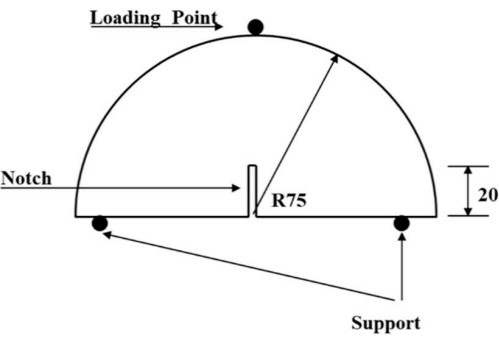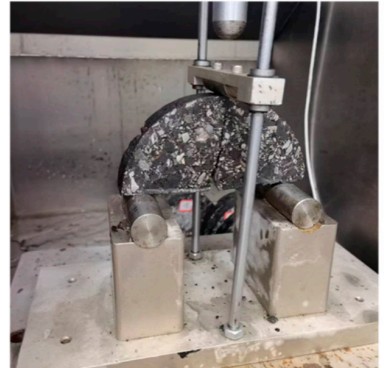

**Figure 6.** Experimental setup.

*4.2. Model Validation*

Based on the above SCB test, the load–displacement curve of the asphalt concrete specimen was obtained. At the same time, the load–displacement curve was also obtained from the CZM-based FE model developed in this paper. Figure 7 shows the comparison between the load–displacement curves from the experiment and the numerical simulation. It could be observed that the numerical results had a similar trend with the experimental data for both −10 °C and 0 °C. The curves gradually climbed to the peak point and then decreased. For the experiment, the measured peak loads and the displacements were 4.07 kN and 0.74 mm under −10 °C, and 3.88 kN and 1.01 mm under 0 °C. While for the numerical simulation, peak loads of 3.99 kN and 3.61 kN were observed for −10 °C and 0 °C, respectively. The corresponding displacements under −10 °C and 0 °C were 0.79 mm and 1.10 mm, respectively. The relative difference in the peak load between the numerical and experimental results was 1.97% at −10 °C and 6.96% at 0 °C. The relative difference in the displacement between the numerical and experimental results was 6.76% at −10 °C and 8.91% at 0 °C. The values of the relative difference show that the accuracy of the CZM-based FE model is acceptable for the cases discussed in this paper. Moreover, for both the experiment and the numerical simulation, a larger peak load and smaller displacement was observed under lower temperature conditions. This means the numerical model reflects a similar trend as the experiment. Therefore, the CZM-based FE model presented in this paper is reliable.

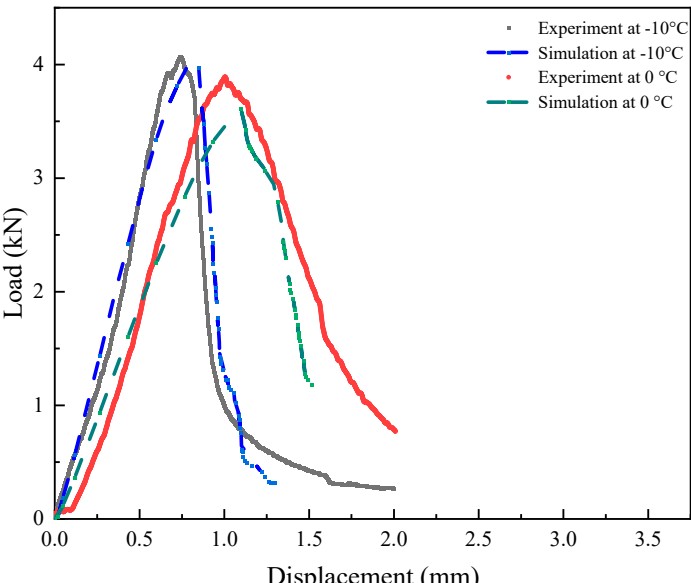

**Figure 7.** Load-Displacement comparison curve.

## 5. Fracture Analysis

In this section, the validated FE model was used to investigate the effects of some critical factors, including temperature, loading rate, aggregate geometry, fracture strength, and fracture energy, on the fracture behavior of asphalt concrete. From the point of view of material composition, the fracture of asphalt concrete may occur in the aggregate, in the asphalt mastic, or at the aggregate–asphalt interface. Previous studies have shown that aggregate only cracks when the temperature is very low [30]. Since this study is mainly focused on the cohesive damage in the asphalt mastic and the adhesive damage at aggregate–asphalt interface, the aggregate damage was ignored.

### 5.1. Fracture Propagation Analysis

Based on the numerical simulation results from this paper, the evolutions of crack length, displacement, and loading during the SCB simulation were obtained. Figure 8 describes the relationship between the crack length and the displacement, as well as the relationship between the load and the displacement.

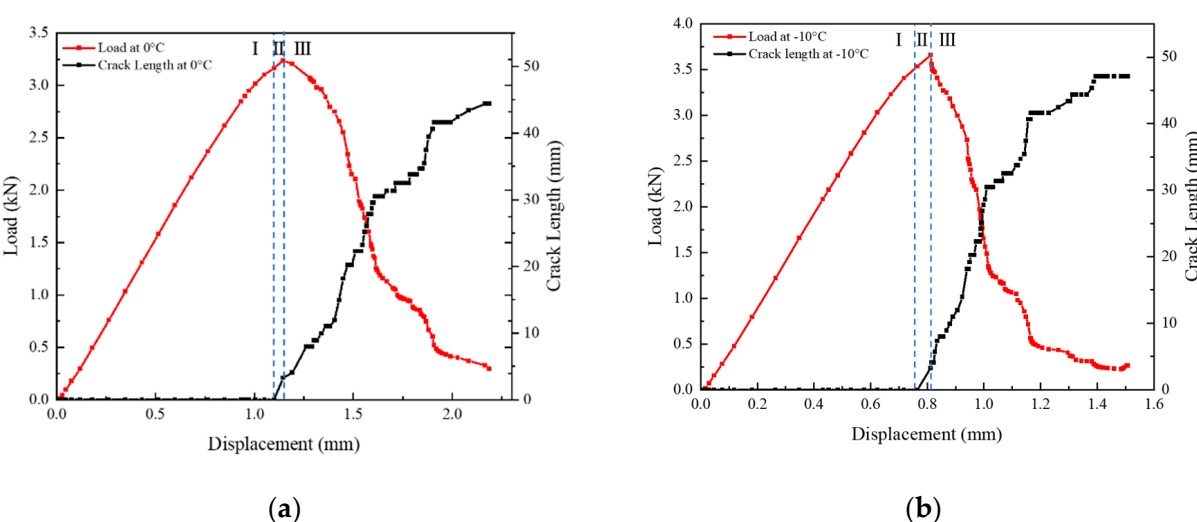

**(a)**　　　　　　　　　　　　　　　　　　　　**(b)**

**Figure 8.** Evolutions of crack length, displacement, and loading during SCB simulation at (**a**) 0 °C; (**b**) −10 °C.

As shown in Figure 8, the fracture process could be divided into three stages from a macro perspective. Stage I was regarded as the crack initiation stage. Stage II represented the crack propagation stage before failure. While stage III was the crack propagation stage after failure. In order to clarify the differences between these stages, the local damage and crack characteristics in these stages at 0 °C were compared in Figure 9 from a mesoscopic perspective.

In Figure 9, the deformation was enlarged three times. The gray elements represent the cohesive damage in asphalt mastic, while the orange elements represent the adhesive damage at the aggregate–asphalt interface. The local damage in Stage I is shown in Figure 9a. It was found that during Stage I, both the gray elements and orange elements were observed near the open end, which means cohesive and adhesive damage occurred due to stress concentration around the open end. However, since stress could still be transmitted through these elements with cohesive and adhesive damage, the load applied to the specimen could still increase.

Figure 9b shows the local damage and cracking in Stage II. It can be seen that some orange elements, representing adhesive damage, turned into a white color, which represents discrete micro-cracks. This means the micro-cracks gradually formed at the aggregate–asphalt interface as the displacement increased. Subsequently, some gray elements, representing cohesive damage, at the tip of the opening changed to a white color, implying the generation of continuous macroscopic cracks. The appearance of cracks further increased

the deformation of the adjacent orange and gray elements, indicating a higher degree of adhesive damage at the nearby aggregate–asphalt interface and cohesive damage in the asphalt mortar. These damages would gradually turn into cracks with increased displacement. The whole specimen would finally reach the maximum load capacity with the growth of cracks. In this process, scalar stiffness degradation (SDEG) was analyzed to describe the damage distribution inside the asphalt concrete specimen. A larger value of SDEG means more serious element damage. In numerical simulations, element damage is judged to be complete when SDEG reaches a critical value [30,50,51]. In this paper, this value was taken as 0.99. Figure 10 shows the distribution of SDEG for cohesive- and adhesive-damaged elements (SDEG > 0) of the entire model at the peak load. The minimum SDEG value of the adhesive elements at the aggregate–asphalt interface was 0.00325, while the minimum SDEG value of the cohesive elements inside the asphalt mortar was 0.00059. The average value of the adhesive elements was 0.441, while the average value of the cohesive elements was 0.424. The damage of the adhesive elements was greater compared to the cohesive elements, indicating that the interface adhesion at the peak load is the weaker part.

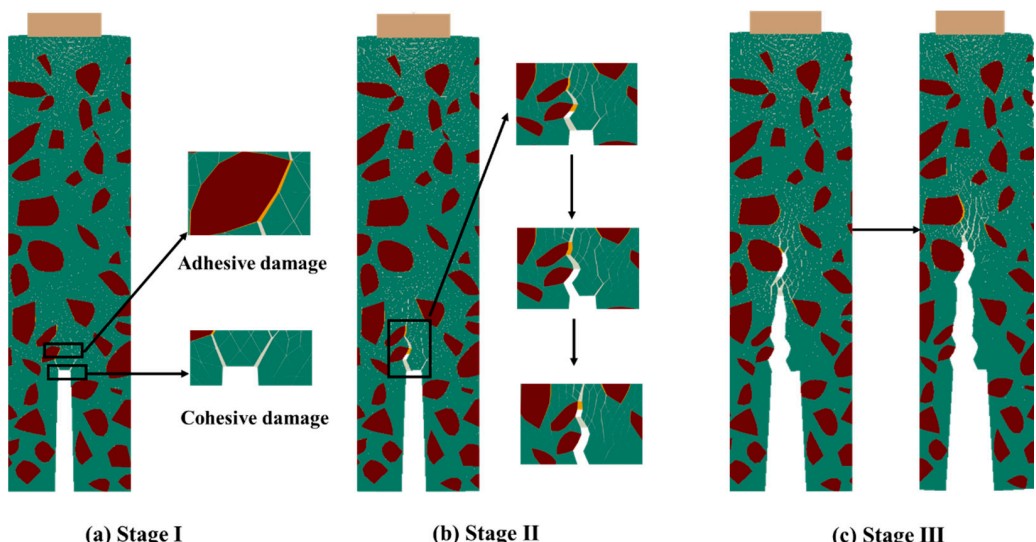

**Figure 9.** Crack propagation process at 0 °C: (**a**) Stage I: Crack initiation; (**b**) Stage II: Crack propagation stage before failure; (**c**) Stage III: Crack propagation.

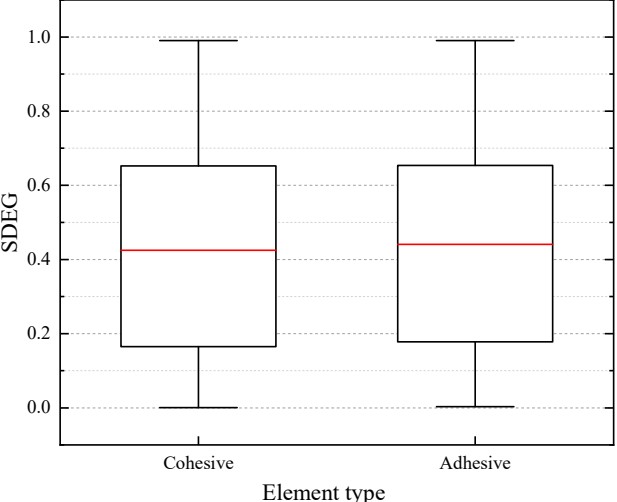

**Figure 10.** SDEG range of damaged elements at peak loads.

In Stage III, the cracks continued to increase, as shown in Figure 9c. Due to the increasing macroscopic cracks, the bearing capacity of the specimen continued to decrease.

### 5.2. Effect of Temperature

As a viscoelastic material, asphalt concrete could exhibit different fracture behavior at different temperatures. In this section, some critical fracture parameters were analyzed with the CZM-based FE model developed in this paper at different temperatures. The loading rate was 5 mm/min in the numerical simulation. Two different temperatures, namely, 0 °C and −10 °C, were considered. The other parameters for numerical simulation were kept the same as that used in model validation. The simulation results showed that asphalt concrete tends to crack earlier at lower temperatures. In detail, for the above simulations, cracks were observed in the specimen when the applied displacement reached 0.812 mm at −10 °C. However, for the same specimen at 0 °C, no cracks were observed until the applied displacement reached 1.141 mm (Figure 8). To find the reason for the above phenomenon, the total number of elements with cohesive damage or adhesive damage under the peak loads were counted. Then the percentage of damaged elements in all elements was calculated and shown in Figure 11.

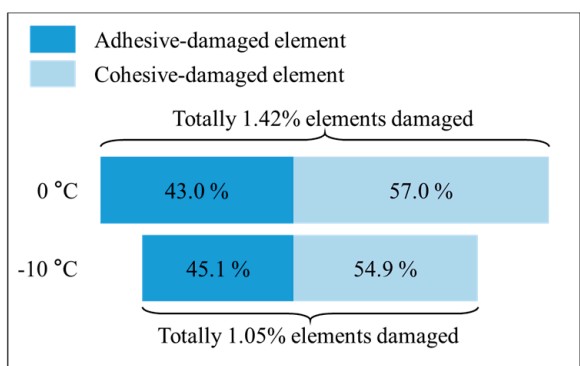

**Figure 11.** Percent of damaged elements at different temperatures.

As shown in Figure 11, although the peak load at −10 °C was higher than that at 0 °C, the cohesive-damaged elements and adhesive-damaged elements at −10 °C were both less than that at 0 °C. The reason for this is that the initial strength of the viscoelastic elements was larger at lower temperatures so that these elements could bear more stress before total failure. Figure 11 also shows that the percentage of adhesive-damaged elements in all damaged elements was higher at lower temperatures, while the percentage of cohesive-damaged elements in all damaged elements was lower at lower temperatures. This indicates that the cracking path might be closer to the aggregates at lower temperatures. However, since the percentage of adhesive-damaged elements only increased by 2.1%, the influence of temperature on the cracking path should not be very significant. This is proved by Figure 12, which shows the simulated crack propagation paths for the same specimen at 0 °C and −10 °C. It can be seen that there is no obvious difference in the crack propagation paths for the same specimen at 0 °C and −10 °C, which indicates that the effects of temperature on crack propagation paths are limited within the temperature range discussed in this paper. This is mainly because the aggregate distribution inside the specimen was the same for the above simulations. As a result, the distributions of the weak parts inside the specimen were almost the same, which tended to induce similar cracking paths. Moreover, the temperatures discussed in this section were not extremely low, so the aggregates were not likely to break.

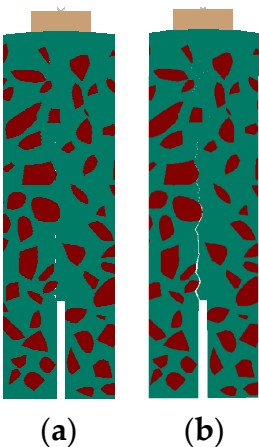

**(a)**　　**(b)**

**Figure 12.** Crack propagation path at (**a**) −10 °C; (**b**) 0 °C.

*5.3. Effect of Loading Rate*

　　Besides temperature, the fracture behavior of viscoelastic asphalt concrete may also be affected by the loading rate. In this section, the effect of loading rate on the fracture behavior of asphalt concrete was analyzed with the CZM-based FE model. In this analysis, the temperature was set to 0 °C. Two different loading rates, namely, 1 and 10 mm/min, were considered. Other parameters for numerical simulation were kept the same as that used in the model validation. Figure 13 shows the total number of damaged elements and the distribution of SDEG values at the peak load under different loading rates. As shown in Figure 13a, more damaged elements were induced at the peak load under higher loading rates. However, further statistical analysis of SDEG values showed that the average value of SDEG was lower under higher loading rates, as shown in Figure 13b. In other words, a higher loading rate may induce more, but minor, element damage. The main reason is that the stiffness of asphalt mortar is larger under a higher loading rate, so the CZM elements in asphalt mortar cannot bear much more stress through deformation. As a result, the excessive stress has to be transferred to adjacent elements and borne by these elements together. In this case, more elements are involved to resist the cracking, with less damage to each single element.

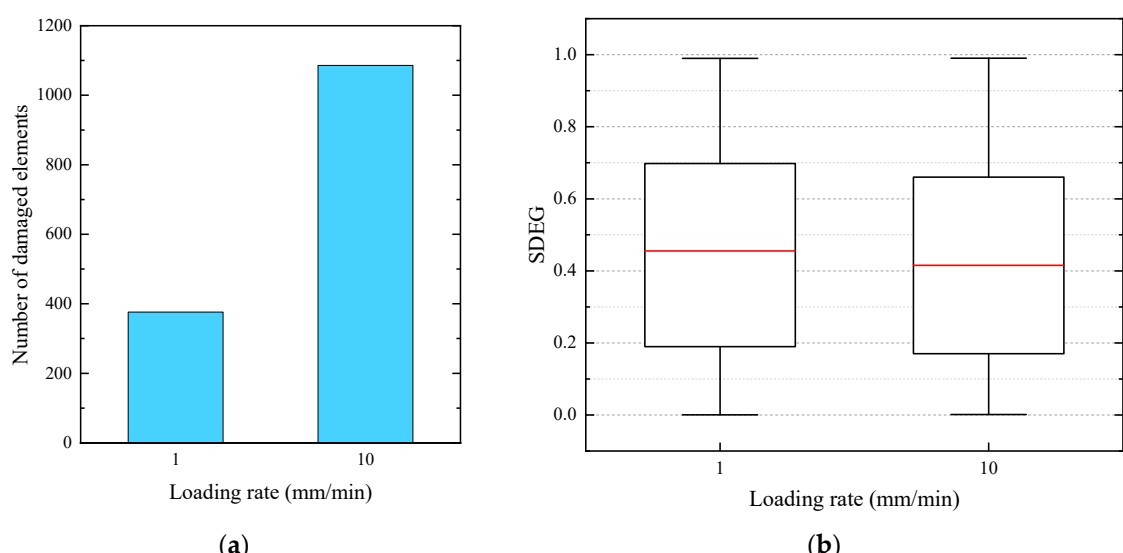

**(a)**　　　　　　　　　　　　　　　　　　　　　　**(b)**

**Figure 13.** Effect of loading rate on fracture behavior: (**a**) total number of damaged elements; (**b**) SDEG distribution.

### 5.4. Effect of Aggregate Geometry

The geometry of aggregates may affect the fracture behavior of asphalt concrete, which emphasizes the importance of aggregate shape in numerical modeling. In order to investigate the effect of aggregate geometry on the fracture behavior of asphalt concrete, two specimens with the same aggregate distribution but different aggregate shapes were generated, as shown in Figure 14. The SCB test was simulated with these specimens under a temperature of 0 °C and a loading rate of 5 mm/min.

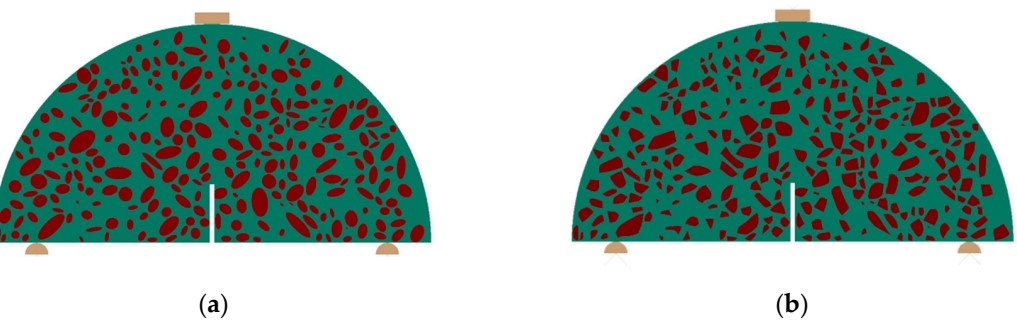

(**a**)                                                                        (**b**)

**Figure 14.** Specimens with the same aggregate distribution but different aggregate shapes: (**a**) ellipsoidal aggregate; (**b**) polyhedral aggregate.

The total number of adhesive-damaged elements and cohesive-damaged elements was counted at the peak loads for the two specimens, and then the percent of damaged elements was calculated, as shown in Figure 15a. It can be seen that the percent of damaged elements for the specimen consisting of ellipsoidal aggregates was lower than that consisting of polyhedral aggregates. At the same time, for the specimen consisting of ellipsoidal aggregates, adhesive-damaged elements only accounted for 37.41% of the total damaged elements. While for the specimen consisting of polyhedral aggregates, this value increased to 43%. In other words, more angular aggregates tend to induce a higher percentage of damaged elements, especially adhesive-damaged elements. This can be further explained by Figure 15b, which shows the distribution of SDEG values for adhesive-damaged elements. As shown in Figure 15b, the maximum values of SDEG in adhesive-damaged elements were 0.96 and 0.99 for the specimen consisting of ellipsoidal aggregates and polyhedral aggregates, respectively. It should be noted that, in the simulation of this paper, any element was treated as failed and turned into a crack when its SDEG value reached 0.99. Therefore, almost no cracks formed at the aggregate–asphalt interface of the specimen consisting of ellipsoidal aggregates, and the crack propagation path should be inside the asphalt mortar phase since the maximum SDEG value of the adhesive element was smaller than 0.99. While for the specimen consisting of polyhedral aggregates, some cracks formed at the aggregate–asphalt interface. This is mainly because the polyhedral aggregates are more angular, which induces stress concentration at the aggregate–asphalt interface.

### 5.5. Effect of Adhesion Strength

As analyzed in the previous sections, stress concentration around angular aggregates may induce excessive stress at the aggregate–asphalt interface. Therefore, the adhesion strength of the interface may have significant effects on the fracture behavior of asphalt concrete. In this section, a series of simulations were conducted to investigate the influence of adhesion strength under a constant temperature of 0 °C and a loading rate of 5 mm/min. The interface adhesion strength was taken as 60%, 100%, and 140% of the initial value used in the model validation. The simulated crack propagation paths are shown in Figure 16. It can be seen that when the interface adhesion strength decreased to 60% of the initial value, local cracks were first observed around aggregates. Then the connection of these local cracks finally formed the main crack propagation path. When the interface adhesion strength increased to 140% of the initial value, the crack only propagated through the asphalt mortar. In general, the interface adhesion strength has an important effect on the

crack propagation path. When the interface adhesion strength is relatively small, the cracks are more likely to propagate around the interface. In this case, the asphalt concrete cracking is controlled by the interface adhesion strength. When the interface adhesion strength is relatively large, the asphalt mortar becomes the weaker part, and the asphalt concrete cracking is controlled by the cohesive strength of the asphalt mortar.

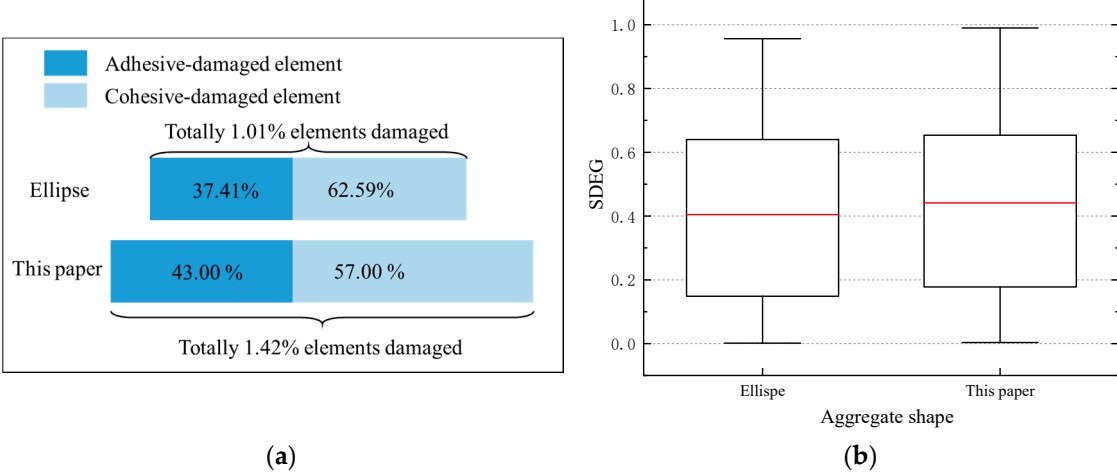

(**a**)　　　　　　　　　　　　　　　　　　　　　　　　(**b**)

**Figure 15.** Effect of aggregate shape on element damage: (**a**) percent of damaged elements; (**b**) SDEG distribution for adhesive-damaged elements.

### 5.6. Effect of Fracture Energy

Fracture energy is an important parameter to describe the fracture behavior of asphalt concrete, which is defined as the amount of energy absorbed to create a unit area of a crack. The material properties of asphalt would affect the fracture energy and further affect the fracture behavior of asphalt concrete. In this section, a series of simulations were conducted to investigate the influence of fracture energy on the fracture behavior of asphalt concrete. The values of fracture energy were taken as 60%, 100%, and 140% of the initial value used in the model validation. The temperature and loading rate during the simulation were 0 °C and 5 mm/min, respectively. Based on the numerical simulation results, the effect of fracture energy on the fracture behavior of asphalt concrete is described in Figure 17. As shown in Figure 17a, when the same displacement was applied, a larger crack length would be induced in asphalt concrete with lower fracture energy. In other words, asphalt concrete tends to crack earlier with lower fracture energy. Figure 17b shows the peak load and displacement in each simulation with different fracture energy. It is observed that with the increase in fracture energy, both the peak load and displacement increase. On average, each 10% increase in fracture energy allows the specimen to bear 2.31% more load and 2.82% more displacement. In general, sufficient fracture energy could improve the ability of asphalt concrete to resist fracture.

### 5.7. Summary of Fracture Analysis

Based on the above analysis, the temperature, loading rate, aggregate geometry, interface adhesive, and fracture energy have an effect on the fracture behavior to some extent. Within the scope of this paper, the effect of temperature on the damage path of asphalt concrete was small. While the loading rate and aggregate geometry had a significant effect on the distribution of damaged elements in the fracture process of asphalt concrete. The change in adhesion and fracture energy directly affected the fracture path and load-carrying capacity of asphalt concrete.

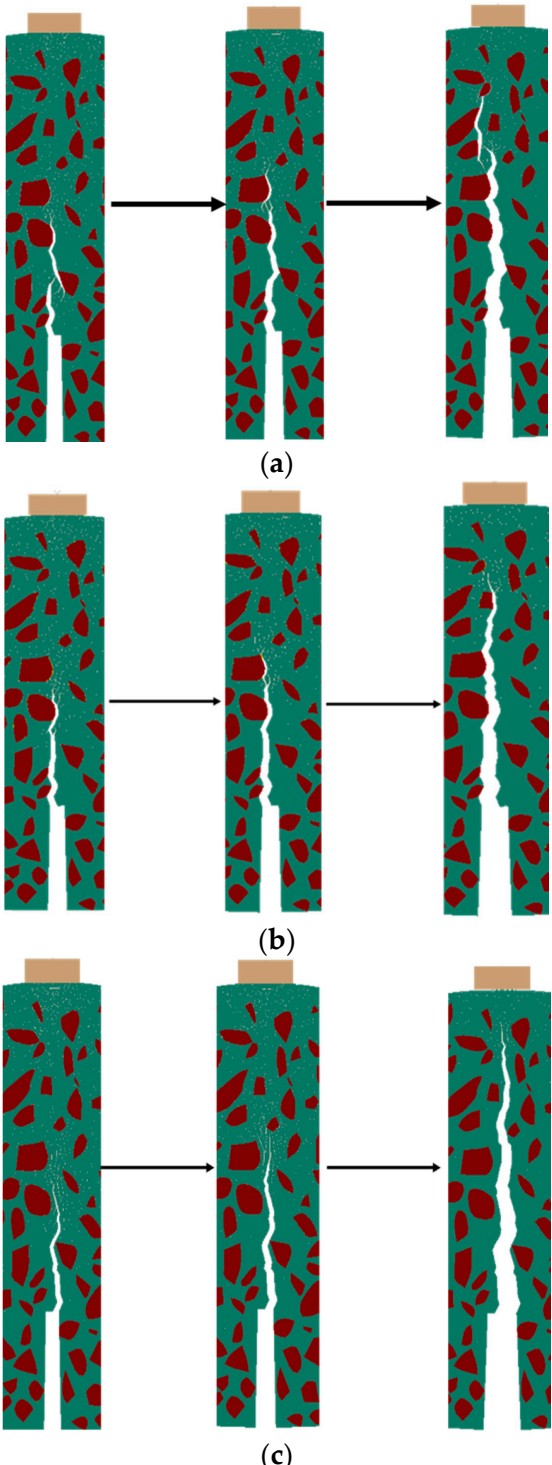

**Figure 16.** Crack propagation path with different interface strengths: (**a**) 60% of initial strength; (**b**) initial strength; (**c**) 140% of initial strength.

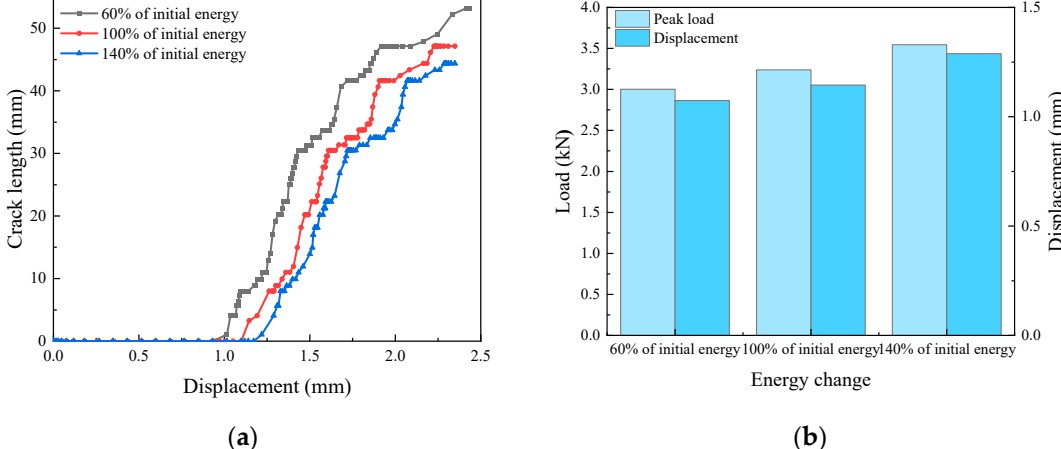

**Figure 17.** Effect of fracture energy on asphalt concrete fracture: (**a**) crack length evolution; (**b**) peak load and displacement.

## 6. Conclusions

In this paper, a CZM-based FE model was developed to evaluate the fracture behavior of asphalt concrete. Two-dimensional models were obtained by cutting from a three-dimensional model based on the real aggregate shape as a way to obtain a numerical model that is closer to the microstructure of the asphalt concrete in the experiment. Numerical simulations of SCB tests were conducted and validated with the experiment. With the validated CZM-based FE model, the effects of some critical factors on the fracture behavior of asphalt concrete were investigated. The following conclusions were obtained:

(1) The damage of the specimens was divided into three stages according to the variation of crack length and load. At the peak load, the average damage of the adhesive elements was higher than that of the cohesive elements, indicating that the aggregate–asphalt interface is the weaker part.

(2) At lower temperatures, asphalt concrete tends to crack earlier, and the cracking path tends to be marginally closer to the aggregates.

(3) Since the stiffness of asphalt mortar is larger under a higher loading rate, the CZM elements in asphalt mortar cannot bear much more stress through deformation. Therefore, a higher loading rate may induce more, but minor, element damage.

(4) Angular aggregates may induce stress concentration at the aggregate–asphalt interface and thus tend to induce a higher percentage of damaged elements, especially adhesive-damaged elements.

(5) On average, each 10% increase in fracture energy allows the specimen to bear 2.31% more load and 2.82% more displacement. Sufficient fracture energy could improve the ability of asphalt concrete to resist fracture.

However, the effects of a wider range of temperatures and the fracture of the aggregates were not considered in this paper. In future studies, a wider range of temperatures and more typical aggregate shape parameters will be considered.

**Author Contributions:** Conceptualization, J.C.; writing—original draft, X.O.; writing—review and editing, X.S. All authors have read and agreed to the published version of the manuscript.

**Funding:** This research was funded by the National Natural Science Foundation of China (Grant No. 51908558 and 52278468), the Natural Science Foundation of Hunan Province (Grant No. 2020JJ5717), and the Hunan Transportation Science and Technology Project (Grant No. 202017).

**Data Availability Statement:** The data presented in this study are available upon request.

**Conflicts of Interest:** The authors declare no conflict of interest.

## Nomenclature

| | | | |
|---|---|---|---|
| CZM | Cohesive zone model | $\delta$ | Separation displacement between interface |
| SCB | Semi-circular bending | $\delta_\text{in}$ | Separation displacement between interface at the initial damage point |
| XFEM | Extended finite element method | $\delta_\text{fin}$ | Separation displacement between interfaces when the cohesive element failed |
| FE | Finite element | $\sigma$ | Cohesive strength |
| $E_\infty$ | Long-term equilibrium relaxation modulus | $\sigma_\text{max}$ | Cohesive strength at the initial damage point |
| $E_\text{i}$ | Relaxation modulus of spring elements in the generalized Maxwell model | $G_\text{I}$ | Fracture energy |
| K | Initial stiffness | SDEG | Scalar stiffness degradation |

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
