# Peer review of "Numerical Investigation of Asphalt Concrete Fracture Based on Heterogeneous Structure and Cohesive Zone Model"

_applsci, doi:10.3390/app122111150_

Round 1

Reviewer 1 Report

The paper must contain a nomenclature - a list of signs and symbols, abbreviations, etc. In the paper, all signs, including those in the tables, must be explained, if it has not been done before.

Table 3 - The unit of stress is "MPa" - not "Mpa". Please correct it. In this table there is the GI value - the energy release factor, which can be equated with the J integral for linear materials. In general, in manuscripts in the field of fracture mechanics, the unit "N/mm" is given here - however, "J/m2" can be left.

Section 3.4. The authors must pay more attention to the description of the numerical model. Indicate the number of nodes in the model, the number of elements, determine whether they assumed the dominance of a plane stress state or a plane strain state. It is worth giving the number of numerical integration points in one finite element. It must be clearly stated whether the notch growth was modeled or the focus was only on the stationary problem. Information must be provided as to whether the contact problem was resolved and how the support and load roller were accurately modeled. It is recommended to provide the dimensions of the finite elements and to add a drawing showing the region at the notch tip to the manuscript.

The text must contain information whether the authors examined the numerical model for convergence, or tested other elements - we are basically bending here, so 2D 8 or 9 nodal elements are recommended.

The paper must include a technical drawing of the sample, with all dimensions. It is necessary.

Moreover, if the increase in notch length was modeled, it should be determined how the increase in its length was controlled.

In Figures - starting with number 7, a space should be added between the axis description and the unit given in round brackets.

The paper is noteworthy, but needs some corrections. Please complete it, have it checked by a professional translation agency and send it for another review.

Reviewer 2 Report

Comments

1.     A comparative analysis needs to include in the study for discussing the novelty of the study while comparing it with previous studies.

2.     In the introduction section, the authors have discussed semi-circular bending (SCB) tests, three-point bending tests, pull-off tests, and direct tensile tests. It is suggested to discuss these tests in tabular form with specifications, parameters, and limitations.

3.     The literature review of the current study is small, it is suggested to improve the literature review with the relevant and latest research.

4.     Before the conclusion section, a new section needs to create for a brief discussion of all the numerical findings that has obtained in the above section for better readability.

5.     The novelty, limitations, and future work of the study need to be discussed in the abstract and conclusion.

Reviewer 3 Report

Asphalt concrete is a composite heterogeneous material, the fracture behaviour of asphalt concrete is affected by its internal composition and meso-structure. Examining the relationship between the internal structure of asphalt concrete and its fracture behaviour can guide the design studies for sustainable and durable pavement. It is aimed to evaluate the fracture behaviour of asphalt concrete by applying semi-circular bending (SCB) test in experimental and numerical simulations. The damage evolution of asphalt concrete can be complex and the crack propagation could be flexible for the extended finite element method (XFEM). Due to these issues, the cohesive zone model (CZM) was preferred. The experiments confirmed the numerical simulations of SCB tests.

The paper is well written and understandable. The content of the research overlaps with the purpose of the research. The subject has been discussed from different aspects.

In addition to supporting numerical simulation with experiments, the study has been strengthened by explaining in detail the factors affecting the result.

Here, my comments for the authors:

  • Why was the bilinear traction-separation law applied to the cohesive elements to achieve damage variation?

  • What is the reason for choosing the element types of aggregates-asphalt mortar and cohesion elements as CPS3 and COH2D4?

  • Why were samples placed in the ambient chamber for more than 4 hours for each test?

  • How are the support positions chosen? If the supports were closer to each other, what kind of change would be observed in the fracture behaviour?

Round 2

Reviewer 1 Report

The authors included all my suggestions in the revised version of the paper. I recommend the manuscript for publication.